

# Adaptive multi-source domain collaborative fine-tuning for transfer learning

Le Feng[1], Yuan Yang[1], Mian Tan[1], Taotao Zeng[1], Huachun Tang[1], Zhiling Li[1], Zhizhong Niu[2] and Fujian Feng[1]

[1] Guizhou Key Laboratory of Pattern Recognition and Intelligent System, Guizhou Minzu University, Guiyang, China

[2] Guizhou University of Commerce, Guiyang, Guizhou, China

## ABSTRACT

Fine-tuning is an important technique in transfer learning that has achieved significant success in tasks that lack training data. However, as it is difficult to extract effective features for single-source domain fine-tuning when the data distribution difference between the source and the target domain is large, we propose a transfer learning framework based on multi-source domain called adaptive multi-source domain collaborative fine-tuning (AMCF) to address this issue. AMCF utilizes multiple source domain models for collaborative fine-tuning, thereby improving the feature extraction capability of model in the target task. Specifically, AMCF employs an adaptive multi-source domain layer selection strategy to customize appropriate layer fine-tuning schemes for the target task among multiple source domain models, aiming to extract more efficient features. Furthermore, a novel multi-source domain collaborative loss function is designed to facilitate the precise extraction of target data features by each source domain model. Simultaneously, it works towards minimizing the output difference among various source domain models, thereby enhancing the adaptability of the source domain model to the target data. In order to validate the effectiveness of AMCF, it is applied to seven public visual classification datasets commonly used in transfer learning, and compared with the most widely used single-source domain fine-tuning methods. Experimental results demonstrate that, in comparison with the existing fine-tuning methods, our method not only enhances the accuracy of feature extraction in the model but also provides precise layer fine-tuning schemes for the target task, thereby significantly improving the fine-tuning performance.

# INTRODUCTION

Transfer learning is an important machine learning technique that aims to utilize knowledge already acquired in the source domain to assist in the learning of related but different target tasks. This method has achieved significant success in a variety of tasks for which training data is insufficient or of poor quality (*Pan & Yang, 2010*; *Zhuang et al., 2021*). Model

Corresponding authors
Mian Tan, tanmian@gzmu.edu.cn
Fujian Feng,
fujian_feng@gzmu.edu.cn

fine-tuning, as a commonly used transfer learning technique, effectively improves the learning performance by transferring the parameters of the pre-trained neural network (NN) model from the source domain to the target task and fine-tuning them (*Rasheed et al., 2023*; *Li et al., 2023a*). Such fine-tuning techniques are already successfully used in various fields such as computer vision (*Dai et al., 2023*), natural language processing (*Chi et al., 2023*), speech recognition (*Chen & Rudnicky, 2023*), recommendation systems (*Liao et al., 2022*), and medical diagnosis (*Li et al., 2023b*).

During the fine-tuning process, the source domain model is usually pre-trained on a large-scale dataset (*e.g.*, ImageNet; *Yamada & Otani, 2022*) and its parameters are replicated in the target model. In order to make the model better adapt to the target task, its parameters are usually fine-tuned to improve its alignment with the target task. However, most of current research focuses on the single-source domain fine-tuning problem, *i.e.,* transferring a single source domain model for a specific target task and fine-tuning it (*Pan & Yang, 2010*), as shown in Fig. 1. When this strategy is adopted, the model performance is primarily determined by the selection of parameters during the fine-tuning process. Accordingly, parameters that are pertinent to the target task should remain unchanged to help extract features common. Parameters that are irrelevant or potentially detrimental to the model's performance should be fine-tuned to enhance the model's adaptive capacity. However, this requires accurate determination of the correlation between the source domain model and the target task, which can be challenging and further complicates the selection of appropriate fine-tuning parameters. Traditional methods for addressing this issue rely on expert experience or trial-and-error methods (*Zhuang et al., 2021*). Some researchers have implemented transfer learning by fine-tuning all layers of the model, known as full or standard fine-tuning (*Dosovitskiy et al., 2021*; *Kornblith, Shlens & Le, 2019*). Although this method is effective in improving the model performance compared to that achieved when the NN model is trained *de novo*, it may lead to overfitting when the source domain model is large and the target dataset is small (*Xuhong, Grandvalet & Davoine, 2018*). To address this problem, some researchers proposed fine-tuning a subset of layers only (*Peters, Ruder & Smith, 2019*; *Xu et al., 2021*; *Shen et al., 2021*). However, even this strategy necessitates expert knowledge to determine which layers require fine-tuning, or this is done by trial-and-error. With the increasing scale of NN models, traditional fine-tuning methods not only become expensive but may also fail to provide precise fine-tuning schemes for the target task. Therefore, solving the problem of adaptive fine-tuning in large-scale models has become an important research focus.

In order to address the issue of large model scale and difficulty in selecting fine-tuning layers, optimization-based fine-tuning methods are proposed to adaptively select the layers that need to be fine-tuned (*Wang, Chen & Ghamisi, 2022*; *Nagae et al., 2022*; *Lee et al., 2023*). This approach reduces the cost of fine-tuning and enhances the alignment between the source domain model and the target task. Optimization-based fine-tuning methods model the problem of selecting fine-tuning layers as an optimization problem with decision variables and iteratively optimizes to find appropriate layer fine-tuning schemes for the target task. Optimization-based methods can be further categorized into policy network-based and evolution optimization-based fine-tuning methods. The

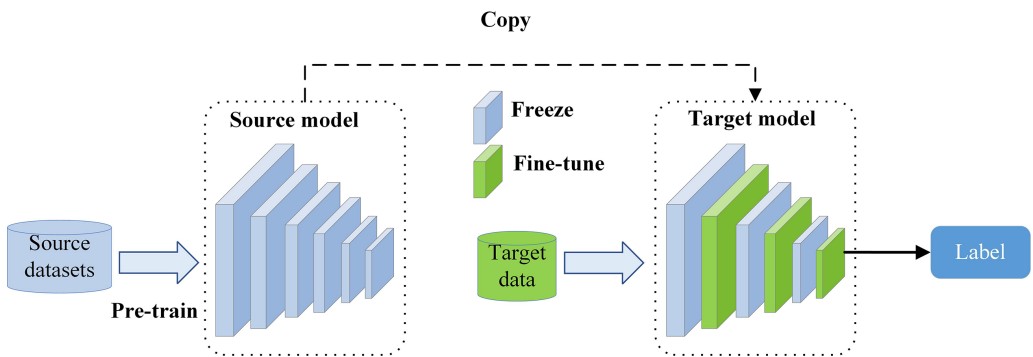

**Figure 1** Graphical representation of the single-source domain fine-tuning process.

former train an additional NN to make binary decisions for each layer in the source domain model, segregating those that need to be fine-tuned from those that do not (*Wang, Chen & Ghamisi, 2022*; *Jang et al., 2019*). While this method quickly provides suitable layer fine-tuning schemes for the target task, the optimization of policy networks based on discrete parameter training requires extensive training support. Therefore, when applying gradient descent directly, optimization can usually only be achieved by estimating approximate gradients. This, in turn, affects the accuracy of the fine-tuning schemes (*Guo et al., 2020*). On the other hand, evolution optimization-based fine-tuning methods consider each fine-tuning scheme as an individual member of a population and find suitable layer fine-tuning schemes for the target task through population evolution (*Nagae et al., 2022*; *Wen, Peng & Ting, 2021*; *de Lima Mendes et al., 2021*). This method avoids gradient computation and enables accurate search for fine-tuning schemes. The aforementioned methods apply to single-source domain model fine-tuning. This method is suitable when the data distribution in the source and target domains is very similar. It helps the model learn the key features of the target task. However, when there is a significant data distribution difference between the source and the target domain, existing single-source domain fine-tuning methods often struggle to extract sufficient key feature information and experience performance degradation.

Spurred by the current deficiencies, this article proposes an adaptive multi-source domain collaboration fine-tuning framework (AMCF). The AMCF aims to address the issue that existing single-source domain fine-tuning methods struggle to extract effective features when there are significant differences in data distribution between the source and target domains. AMCF combines multiple source domain models for fine-tuning to enhance the ability of the model to extract key features from the target task. To determine the appropriate fine-tuning layer for multiple source domain models, an adaptive multi-source domain layer selection strategy (AMLS) is designed. AMLS dynamically selects the layer for fine-tuning and freezing in each source domain model based on the disparity between the target task and the source task, allowing the model to extract more key feature information. Moreover, to address the challenge posed by the variations among multiple source domain models, which hinder the extraction of effective features for the target task,

a multi-source domain collaborative loss function (MC-loss) is proposed. The objective of MC-loss is to minimize the disparities in the outputs of the various source domain models, thereby enhancing the accuracy of the features extracted by the models for the target task. The main contributions of this work are summarized below:

(1) We propose an adaptive multi-source domain collaborative fine-tuning framework, which effectively enhances the feature extraction capability of the model through collaborative fine-tuning of multiple source domain models. This is in contrast to the current fine-tuning methods that mainly focus on fine-tuning a single source domain model to the target domain, making it challenging to extract essential feature information for the target task when there are significant differences in data distribution between the source and target domains.

(2) We also propose an adaptive multi-source domain layer selection strategy that dynamically provides suitable layer fine-tuning schemes for the target task using multiple source domain models, enabling the model to extract more essential feature information.

(3) A multi-source domain collaborative loss function has been proposed, which aims to enhance the adaptability of each source domain model to the target task while minimizing the disparities between outputs of different source domain models, enabling the model to extract key feature information more accurately.

The remainder of this article is presented in four sections. 'Related work' covers related work. 'Adaptive multi-source domain collaborative fine-tuning framework' introduces the adaptive multi-source domain collaborative fine-tuning framework, which includes the adaptive multi-source domain layer selection strategy and the multi-source domain collaborative loss function. The experiments carried out on publicly available visual classification datasets to verify the utility of the proposed framework are presented in 'Experimental design and analysis of the obtained results'. This article concludes with 'Conclusion', and offer suggestions for future work in this domain.

## RELATED WORK

Recently, the field of fine-tuning has experienced significant advancements, as evident from numerous publications that can be broadly categorized into those related to traditional methods and optimization-based methods. In the brief literature review provided below, we will focus on the work most closely related to this article.

### Traditional fine-tuning methods

During the fine-tuning process, the selection of fine-tuning layers directly impacts the model's performance on the target task (*Yosinski et al., 2014*). Traditional fine-tuning methods focus on determining which layers need to be fine-tuned through expert experience and manual trial-and-error. In this regard, some researchers have chosen to fine-tune all layers for training on the target task, an method known as full fine-tuning or standard fine-tuning (*Chen et al., 2020*; *Raffel et al., 2020*). For example, *Dosovitskiy et al. (2021)* transferred a pre-trained Visual Transformer (ViT) model to multiple small-scale datasets and fine-tuned all the layers with good results. Moreover, *Raghu et al. (2020)* transferred a pre-trained source domain model to the task of classifying

epileptic seizure types from available electroencephalograms and fine-tuned all the model's layer parameters, which significantly improved its performance. As a part of their work, *Kornblith, Shlens & Le (2019)* conducted extensive experiments and determined that the source domain model accuracy when applied to the source task is strongly correlated with the accuracy transferred to the target task, highlighting the significance of source domain knowledge for the target task. In order to enhance the stability of fine-tuning, *Qiu et al. (2023)* proposed an Orthogonal Fine-Tuning (OFT) method, which employs an orthogonal transformation to fine-tune model parameters, enabling the pre-trained model to adapt to downstream tasks without altering the model's hyperspherical energy. However, when the amount of data related to the target task is small and the source domain model is large, the standard fine-tuning approach easily lead to overfitting.

To address the issue of overfitting during model fine-tuning, some researchers have opted for freezing specific layers and fine-tuning only a subset of the model. For example, *Basha et al. (2021)* effectively improved model generalizability to multiple image classification tasks by fine-tuning the last few layers of the source domain model and freezing the remaining layers. However, this approach still requires manual setting of the number of fine-tuned layers. To address this issue, *Zunair, Mohammed & Momen (2018)* utilized a transfer learning technique and applied it to the Bengali digit classification problem. After conducting several experiments, these authors demonstrated that the best classification accuracy can be attained by selecting the first input layer and the last fully connected layer of the source domain model as the fine-tuning layers. Similarly, *Ghafoorian et al. (2017)* achieved the best performance in a transfer learning task for magnetic resonance imaging (MRI) segmentation by fine-tuning only the first six layers of the source domain model. While these results are encouraging, as the size of convolutional neural networks (CNNs) increases, relying on manual human trial-and-error input to determine the layer parameters that need to be fine-tuned becomes time consuming and costly. Therefore, there is a need to develop automated methods for accurately selecting the layers to be fine-tuned.

**Optimization-based fine-tuning methods**

Optimization-based fine-tuning is an adaptive parameter selection method that models the fine-tuning layer selection as a combinatorial optimization problem which is solved through iterative optimization. Currently available methods based on this approach can be categorized into policy network-based methods and evolutionary optimization-based methods. The former treat the decision of whether to fine-tune each layer parameter of the source domain model as a binary variable and rely on an additional NN to determine a suitable fine-tuning scheme for the target task (*Jang et al., 2019*; *Guo et al., 2019*; *Chen et al., 2023*). In their work focusing on image classification tasks, *Guo et al. (2019)* utilized a recurrent gate network to make binary decisions on fine-tuning each layer during the fine-tuning process. On the other hand, *Chen et al. (2023)* introduced a user-specific adaptive fine-tuning method (UAF) that determines which layers of the source domain model to fine-tune for each input, whereby layer selection was accomplished using an additional policy network. To enhance model performance, *Jang et al. (2019)* proposed

a transfer learning method based on meta-learning techniques to adaptively determine which layers should be fine-tuned or frozen. Policy network-based methods are superior to traditional fine-tuning methods as they can quickly predict fine-tuning schemes suitable for the target task. While these approaches not only reduce costs but also significantly enhance the performance of fine-tuned models, this necessitates training a large number of discrete parameters. Moreover, due to the discreteness and non-differentiability of the fine-tuning parameter selection problem, directly applying gradient descent may affect the accuracy of the fine-tuning scheme (*Guo et al., 2019*).

These issues can be mitigated by applying the evolutionary optimization-based fine-tuning method, as it treats each fine-tuning scheme as an individual in the population. It searches for the best fine-tuning scheme through operations such as iterative population selection, crossover, and mutation. *Vrbančič & Podgorelec (2020)* used the differential evolution algorithm to solve the combinatorial optimization problem of selecting the fine-tuning layer and determine the optimal fine-tuning scheme. *Wen, Peng & Ting (2021)* proposed a two-stage evolutionary transfer learning method to address the challenge of adapting the model structure to the target task in transfer learning. This method optimizes the selection problem of the fine-tuning layer by applying multi-objective optimization during the fine-tuning stage. Genetic algorithms have also been utilized to address the fine-tuning layer selection problem. For instance, they were adopted by *Nagae et al. (2022)* and *de Lima Mendes et al. (2021)* to identify effective layers suitable for fine-tuning, whereby the fine-tuning scheme was optimized using different initialization and crossover strategies. Additionally, *Hasana, Ibrahim & Ali (2023)* developed a genetic algorithm to select modules for fine-tuning in source domain models that are typically composed of multiple modules. As shown above, the evolutionary optimization-based methods eliminate the need for gradient computation and can accurately identify fine-tuning schemes that are suitable for the target task.

However, the above methods are only suitable for fine-tuning a single source domain when the distribution difference between the source domain and the target domain is small. To extract more effective features for the target task, our work mainly focuses on collaborative fine-tuning of multiple source domain models, given that the learning performance when the method is applied to the target task can be further improved by jointly fine-tuning multiple source domain models. The following section will introduce the proposed adaptive multi-source domain collaborative fine-tuning framework in detail.

# ADAPTIVE MULTI-SOURCE DOMAIN COLLABORATIVE FINE-TUNING FRAMEWORK

## Overview of the proposed framework

Most existing fine-tuning methods focus on fine-tuning individual source domain models to the target domain. However, accurately extracting effective features from the target domain can be challenging when there is a significant difference in data distribution between the source and the target domain (the large difference in the feature distribution

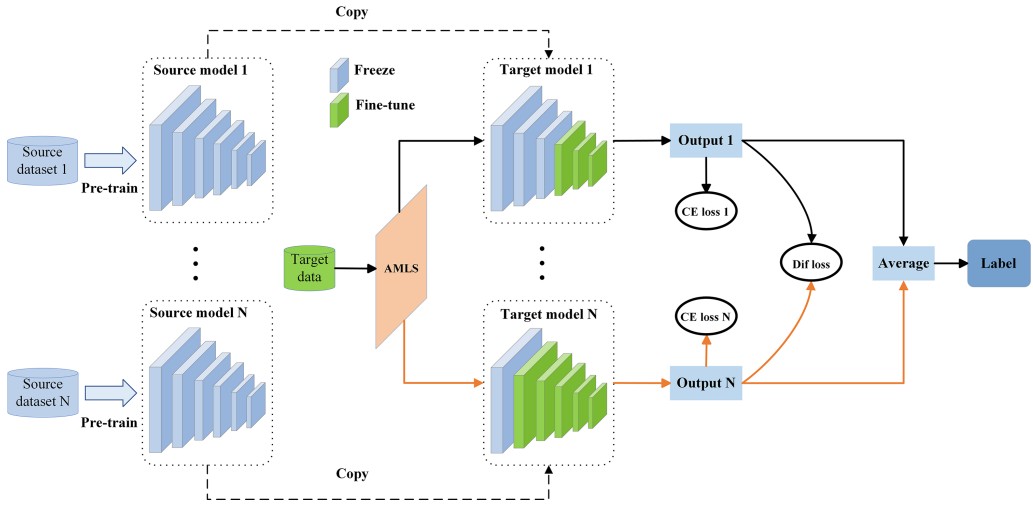

**Figure 2** Framework diagram of the proposed adaptive multi-source domain collaborative fine-tuning (AMCF), where "AMLS" represents the proposed adaptive multi-source domain layer selection strategy, "CE loss" represents the cross-entropy loss function, and "Dif loss" represents the difference loss function.

and category label distribution between the source dataset and the target data). We address this issue by proposing an adaptive multi-source domain collaborative fine-tuning framework (AMCF) for collaborative fine-tuning of multiple source domain models, depicted in Fig. 2. AMCF conducts collaborative fine-tuning through multiple source domains. Compared to using a single source domain, this approach increases the chances for the model to provide more feature information for downstream target tasks. For example, source domain model 1 may lack certain key feature information of the target task (with significant distribution differences), but source domain model 2 may contain such information. Therefore, the proposed method aggregates this information to help learn better features and improve model performance. The AMCF can dynamically determine which layers should be fine-tuned and frozen for multiple source domain models. The layers chosen for fine-tuning are updated during training, while the layers chosen for freezing retain their parameter values. The selection of fine-tuning layers is accomplished *via* an adaptive multi-source domain layer selection strategy (AMLS), whereby the target data is processed by each source domain model to produce an output value. To reduce the discrepancies in output values yielded by different source domain models, we take the average of $N$ output values to obtain the predicted label. In fact, the core principle of the AMCF lies in the collaborative fine-tuning between different source domain models. By utilizing multiple source domain models, it is possible to extract more comprehensive features that can aid in the training of the target domain. Furthermore, by computing the difference loss between various source domain models, the features extracted by the model can be more precise, thus enhancing the model's performance on the target task.

The AMCF consists of two main components: (1) an adaptive multi-source domain layer selection strategy (AMLS); (2) a multi-source collaborative loss function (MC-loss).

## Adaptive multi-source domain layer selection strategy

During the fine-tuning process, certain layers in the source domain model can extract features that are shared between the source and the target domain. These helpful network layers for the target task should be frozen to enhance generalization ability, while the network layers that are not beneficial for the target task should be fine-tuned to help the model adapt to the target task. Therefore, it is crucial to choose the right layers for fine-tuning in order to achieve optimal model performance. However, due to the large number of source domain models, each model may have dozens or even hundreds of layers, making it difficult to efficiently select the layers that need to be fine-tuned. To address this problem, we propose an adaptive multi-source domain layer selection strategy (AMLS) that can automatically search for appropriate layer fine-tuning schemes from the vast model layers based on the differences between the target and the source domain. By employing AMLS, we can effectively identify the layers that require fine-tuning, rather than indiscriminately fine-tuning all layers, thereby enhancing the model performance on the target task.

We assume that the layers near the initial layer of the source domain model help extract shared features between the source domain and the target domain. Accordingly, we keep these layers frozen, while considering the layers closer to the final layer of the source domain model (which are more susceptible to specific tasks) for fine-tuning. This is accomplished by combining the characteristics of individual models and utilizing the particle swarm optimization (PSO) algorithm to search for the optimal number of fine-tuning layers for multiple source domain models. PSO is currently one of the most popular metaheuristic algorithms, as it is simple to implement and has strong global search capabilities (*Zeng et al., 2022*). When selecting fine-tuning layers for multiple source domain models, we assume that there are $N$ such models, denoted as $\{W^{(1)}, W^{(2)}, \ldots, W^{(N)}\}$. Let $W^{(j)} = \{W_1^{(j)}, W_2^{(j)}, \ldots, W_{\rho_j}^{(j)}\}$ represents the $j$th source domain model, and $W_k^{(j)}$ represents the $k$th layer parameters of the model $W^{(j)}$. $\rho_j$ represents the total number of layers in model $W^{(j)}$. The encoding form of the individuals (candidate solutions) in the problem of selecting fine-tuning layers for multiple source domain models can be described as follows:

$$x_i = (x_{i,1}, x_{i,2}, \ldots, x_{i,N}) \tag{1}$$

where $x_i$ represents the $i$th individual in population, as well as the fine-tuning scheme for the corresponding layers of $N$ models. $x_{i,j}$ represents the number of fine-tuned layers in the $j$th source domain model, and $x_{i,j} \in \{0, 1, \ldots, \rho_j\}$. Moreover, $x_{i,j} = \rho_j$ indicates that all layers of $W^{(j)}$ are fine-tuned. Conversely, $x_{i,j} = 0$ indicates that all layers of $W^{(j)}$ are frozen. In order to evaluate the quality of each individual, we use the classification accuracy of the target task corresponding to individual $x_i$ as the fitness value. Specifically, the fitness function $f$ for individual $x_i$ is represented as follows:

$$f(x_i) = Acc(\{\hat{W}^{(1)}, \hat{W}^{(2)}, \ldots, \hat{W}^{(N)}\}; D_t) \tag{2}$$

where

$$\hat{W}^{(j)} = W^{(j)} - \eta \frac{\partial \mathcal{L}_{total}}{\partial W^{(j)}} \odot M^{(j)}, j = 1, 2, \ldots, N \tag{3}$$

$$M_k^{(j)} = \begin{cases} 0, k \le \rho_j - x_{i,j}, k = 1, 2, \ldots, \rho_j \\ 1, other \end{cases} \tag{4}$$

where $M^{(j)} = \{M_1^{(j)}, M_2^{(j)}, \ldots, M_{\rho_j}^{(j)}\}$ represents a mask matrix that corresponds one-to-one with the parameters dimensions of model $W^{(j)}$, which is mainly used to freeze or fine-tune parameters; $\hat{W}^{(j)}$ denotes the model fine-tuned by the source domain model $W^{(j)}$, using the mask matrix $M^{(j)}$; and $M_k^{(j)}$ represents the mask value corresponding to all parameters $W_k^{(j)}$ in the $k$th layer of the source domain model $W^{(j)}$. When $M_k^{(j)} = 0$, it indicates that the parameters in this layer are frozen (*i.e.*, not being fine-tuned), whereas $M_k^{(j)} = 1$ indicates that the parameters in this layer need to be fine-tuned to adapt to the target task. The value of $M_k^{(j)}$ is determined by the individual $x_i$. $\mathcal{L}_{total}$ represents the multi-source domain collaborative loss function (MC-loss) proposed in this article (as discussed in the next section). $Acc(\{\hat{W}^{(1)}, \hat{W}^{(2)}, \ldots, \hat{W}^{(N)}\}; D_t)$ represents the classification accuracy of the target task $D_t$, which is obtained by fine-tuning the corresponding layer parameters based on individual $x_i$ in the framework of Fig. 2. To reduce computational cost, the number of fine-tuning epochs for each individual is manually set to the *maxEpoch* (in this case, it is set to 5). Overall, our goal is to search for a individual $x^*$ that maximizes the fitness value as much as possible.

In the PSO, each particle has its own position (candidate solution) $x_i$ and velocity $v_i$, which are continuously updated based on the guidance of individual historical best and the global best of the population, respectively, by applying the following expressions:

$$v_{i,d}^{(t+1)} = \alpha * v_{i,d}^{(t)} + c_1 r_1 (p_{i,d} - x_{i,d}^{(t)}) + c_2 r_2 (p_{g,d} - x_{i,d}^{(t)}) \tag{5}$$

$$x_{i,d}^{(t+1)} = x_{i,d}^{(t)} + v_{i,d}^{(t+1)} \tag{6}$$

where $x_{i,d}^{(t)}$ and $v_{i,d}^{(t)}$ denote the position and velocity, respectively, of the $i$th individual in the $t$th generation population within the $d$th dimension; $p_{i,d}$ and $p_{g,d}$ respectively represent the historical optimal position of the $i$th individual on the $d$th dimension and the global optimal position of the population; $r_1$ and $r_2$ are random numbers ranging from 0 to 1; $\alpha$ denotes the inertia weight; $c_1$ and $c_2$ represent the individual learning factors.

---

**Algorithm 1** : Adaptive multi-source domain layer selection strategy.

---

**Input:** The source domain models $W^{(1)}, W^{(2)}, \ldots, W^{(N)}$; target data $D_t$; population size $n$; the max iteration number $G$ of the population.

1. Randomly generate populations with $n$ individuals: $(v_i, x_i), i = 1, \ldots, n$.

2. Initialize the individual optimal solution: $p_i = x_i, i = 1, \ldots, n$; the population optimal solution:

   $p_g = x_1$.

3. **for** $i = 1$ **to** $G$ **do**

4.     **for** $j = 1$ **to** $n$ **do**

5.          **for** $k = 1$ **to** $N$ **do**

6.              Calculate mask matrix $M^{(k)}$ according to $x_{j,k}$ using Eq. (4).

7.              Fine-tune model $W^{(k)}$ *maxEpoch* epochs on target data $D_t$ using Eq. (3).

8.          **end for**

9.          Calculate fitness value $\varphi_j$ of $j$th individual $x_j$ using Eq. (2).

10.        **if** $\varphi_j$ better than $p_j$ **then**

11.           $p_j = x_j$.

12.           **if** $\varphi_j$ better than $p_g$ **then**

13.              $p_g = x_j$.

14.           **end if**

15.        **end if**

16.     **end for**

17.     $x^* = p_g$.

18.     Update all individual velocity $v$ values using Eq. (5).

19.     Update all individual position $x$ values using Eq. (6).

20.**end for**

**Output:** the optimal layer fine-tuning scheme $x^*$.

---

The AMLS is described in Algorithm 1. As can be seen from the presented steps, we first randomly generate a population with $n$ individuals and initialize the velocity $v = \{v_i, i = 1, \ldots, n\}$ and the position $x = \{x_i, i = 1, \ldots, n\}$ (line 1). Each individual's position $x_i$ represents the layer fine-tuning scheme corresponding to $N$ source domain models. Next, the individual best solution $p_i, i = 1, \ldots, n$ and the population best solution $p_g$ of $n$ individuals are initialized respectively (line 2). Then, the masking matrix $M^{(k)}, k = 1, \ldots, N$ corresponding to $N$ source domain models is calculated based on each individual's position information (lines 6–8), and the $N$ source domain models $\{W^{(1)}, W^{(1)}, \ldots, W^{(N)}\}$ are fine-tuned *maxEpoch* times using the MC-loss function. The fitness value of each individual is calculated separately using Eq. (2), and the individual optimal position $p_j$ and the population optimal position $p_g$ are updated (lines 9–13). The current population optimal position $p_g$ is assigned to the optimal layer fine-tuning scheme $x^*$, and the individuals' velocities and positions are updated using Eqs. (5) and (6). After $G$ iterations of updates, the optimal layer fine-tuning scheme $x^*$ for $N$ source domain models is obtained.

The time complexity of Algorithm 1 is primarily determined by the population size $n$, the maximum number of iterations $G$, the number of source domain models $N$, and the number of fine-tuning epochs *maxEpoch*. Initializing the particle swarm step requires setting a fine-tuning state for each layer of each model, resulting in a time complexity of $O(n * N)$. During the fitness evaluation step, each particle's performance is evaluated on the target dataset, requiring fine-tuning each model for *maxEpoch* epochs. So the time complexity of this part is $O(n * G * N * maxEpoch)$ because fitness evaluation is needed in each iteration. The update of particle positions and velocities step has a time complexity of $O(n * G * N)$, as each particle's position and velocity need to be updated in every iteration. Thus, the total time complexity of Algorithm 1 is the sum of the above steps, which can be simplified to $O(n * G * N * maxEpoch)$. This complexity indicates that the time complexity of Algorithm 1 primarily depends on the population size, the maximum number of iterations, the number of models, and the number of fine-tuning epochs.

## Multi-source domain collaborative loss function

Based on the AMLS, we can fine-tune the corresponding layers in multiple source domain models. However, when the differences between multiple source domain models are too large, it becomes challenging for the model to accurately extract key feature information for the target task. Therefore, inspired by the multi-source domain adaptation method (*Zhu, Zhuang & Wang, 2019*), we propose a multi-source domain collaborative loss (MC-loss) function to enhance the capability of multiple source domain models in extracting crucial features for the target task. Different from the previously reported multi-source domain adaptation methods, the method proposed in this article does not require source domain data, as it solely relies on the pre-trained source domain models, making its application more efficient.

In this article, MC-loss function mainly consists of two parts: cross-entropy (CE) loss and difference (Dif) loss. We utilize cross-entropy loss to enhance the accuracy of feature extraction by multiple source domain models for the target task. However, when there are significant differences between multiple source domain models, using only cross-entropy loss is not sufficient. As shown in the left half of Fig. 3, the features extracted from the three source domain models are feature 1, feature 2, and feature $N$ (corresponding to output 1, output 2, and output $N$ in Fig. 2). Among these features, only feature 2 is good, while the others are poor. Therefore, the extracted features in the end will also be influenced by these poor features. To address this issue, we design a difference loss function that aims to minimize the discrepancy between the outputs of various source domain models. By minimizing the difference loss, poor features (such as feature 1 and feature $N$) will gradually approach superior features (such as feature 2), as shown in the right half of Fig. 3. It is worth noting that due to the previous cross-entropy loss, good features will not move closer to bad features at this time. Therefore, the extracted features will gradually converge to excellent features in the end. The resulting multi-source domain collaborative loss function–a combination of the cross-entropy loss and the difference loss–can be

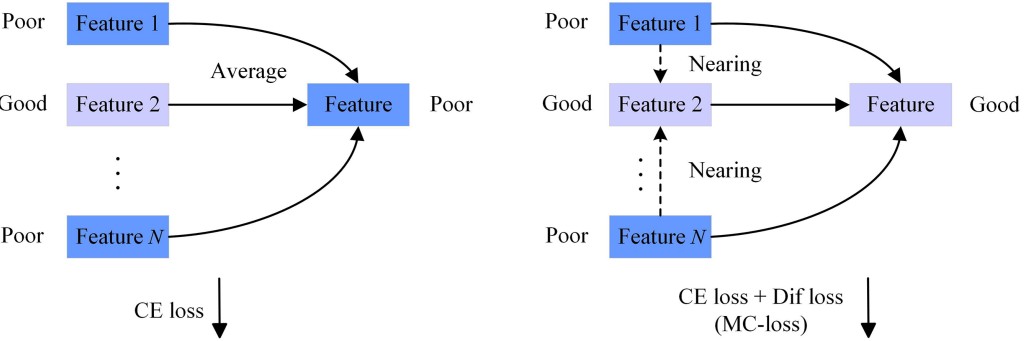

**Figure 3** **The role of the multi-source domain collaborative loss function.**

mathematically represented as follows:

$$\mathcal{L}_{total} = \sum_{k=1}^{N} \mathcal{L}_{CE,k} + \beta \mathcal{L}_{Dif} \tag{7}$$

where

$$\mathcal{L}_{CE,k} = -\frac{1}{|D_t|} \sum_{i=1}^{|D_t|} \sum_{j=1}^{C} y_{i,j} \log O_{i,j}^k \tag{8}$$

$$\mathcal{L}_{Dif} = \frac{1}{|D_t|N} \sum_{i=1}^{|D_t|} \sum_{k=1}^{N} \left\| O_i^k - \bar{O}_i \right\|_2^2 \tag{9}$$

where $\mathcal{L}_{CE,k}$ represents the cross-entropy loss of model $W^{(k)}$ on the target dataset $D_t$; $O_{i,j}^k$ denotes the output value of the $j$th dimension of the $i$th sample in model $W^{(k)}$; $y_{i,j}$ represents the true label of the $i$th sample belonging to the $j$th class; $C$ is the total number of classes in the target dataset; $|D_t|$ represents the total number of samples in the target task; $\beta$ is a hyperparameter; $\|*\|_2^2$ represents the 2-norm of $*$; $\mathcal{L}_{Dif}$ denotes the difference loss between the outputs of $N$ source domain models; $O_i^k$ represents the output of the $i$th sample on model $W^{(k)}$; and $\bar{O}_i$ represents the average value of the outputs obtained by the $i$th sample in $N$ source domain models.

In summary, our total loss function consists of the $\mathcal{L}_{CE,k}$ and $\mathcal{L}_{Dif}$ loss functions. Reducing the $\mathcal{L}_{CE,k}$ loss will improve the accuracy of feature extraction from the source domain models on the target data. However, due to significant differences between the source domain models, those with lower accuracy will impact the averaged predicted labels. Building on the $\mathcal{L}_{CE,k}$ loss, by reducing the $\mathcal{L}_{Dif}$ loss, these lower accuracy models will gradually converge towards the higher accuracy models, significantly improving the accuracy of the final averaged predicted labels.

# EXPERIMENTAL DESIGN AND ANALYSIS OF THE OBTAINED RESULTS

In order to verify the effectiveness of the proposed AMCF in solving the parameter fine-tuning problem in transfer learning, three sets of experiments were conducted. The objective of these experiments was to validate the AMLS, the MC-loss function, and the overall effectiveness of the proposed method, respectively.

## Experimental setup

In order to evaluate the effectiveness of the proposed AMCF, we conducted extensive experiments on seven public datasets and compared the results with those obtained *via* currently utilized fine-tuning methods. In addition, we analyzed the effectiveness of the proposed AMLS and MC-loss function. The following four sets of experiments were performed for this purpose:

(1) Parameter sensitivity analysis and multi-source domain model selection analysis. The aim was to analyze the impact of different hyperparameter values and different source domain selections on the fine-tuning performance of AMCF.

(2) Experiment for establishing the effectiveness of the AMLS. The aim was to verify that the AMLS can accurately identify the layer fine-tuning scheme suitable for the target task.

(3) Experiment for establishing the effectiveness of MC-loss function. Here, the goal was to verify that the MC-loss function can help multiple source domain models learn a greater number of effective features pertaining to the target task.

(4) Experiment for comparing the AMCF proposed here with the current popular single-source domain fine-tuning methods, including Train-From-Scratch, Standard Fine-Tuning (*Kornblith, Shlens & Le, 2019*), L2-SP (*Xuhong, Grandvalet & Davoine, 2018*), Child-Tuning (*Xu et al., 2021*), AdaFilter (*Guo et al., 2020*), ALS (*Nagae et al., 2022*), Auto-RGN (*Lee et al., 2023*), and OFT (*Qiu et al., 2023*).

**Methods used in comparison:** For clarity, a brief description of each method is given below.

- Train-From-Scratch: This method trains the model anew using randomly initialized weights without applying any transfer learning methods.
- Standard Fine-Tuning (Standard FT): This method fine-tunes all the parameters of the source domain model on the target dataset (*Kornblith, Shlens & Le, 2019*).
- L2-SP: This regularized fine-tuning method uses an L2 penalty in the loss function to ensure that the fine-tuned model is similar to the pre-trained model (*Xuhong, Grandvalet & Davoine, 2018*).
- Child-Tuning: This method selects a child network in the source domain model for fine-tuning through a Bernoulli distribution (*Xu et al., 2021*).
- AdaFilter: This method uses a policy network to determine which filter parameters need to be fine-tuned (*Guo et al., 2020*).
- Automatic layer selection (ALS): This method uses a genetic algorithm to automatically select an effective update layer for transfer learning (*Nagae et al., 2022*).

**Table 1  Datasets used to evaluate our method and compare it to other fine-tuning approaches.**

| Target datasets | Training instances | Evaluation instances | Classes |
|---|---|---|---|
| Stanford dogs | 12,000 | 8,580 | 120 |
| MIT indoor | 5,360 | 1,340 | 67 |
| Caltech256-30 | 7,680 | 5,120 | 256 |
| Caltech256-60 | 15,360 | 5,120 | 256 |
| Aircraft | 3,334 | 3,333 | 100 |
| UCF-101 | 7,629 | 1,908 | 101 |
| Omniglot | 19,476 | 6,492 | 1,623 |

- Auto-RGN: The method automatically selects layers with relatively high relative gradient norms for fine-tuning at each epoch (*Lee et al., 2023*).
- Orthogonal Finetuning (OFT): The method employs an orthogonal transformation to fine-tune model parameters, enabling the pre-trained model to adapt to downstream tasks without altering the model's hyperspherical energy (*Qiu et al., 2023*).

**Datasets and the pre-trained models:** For the comparisons, the two most widely used datasets–CIFAR-100 (*Krizhevsky & Hinton, 2009*), ImageNet (*Yamada & Otani, 2022*), and CUB-200 (*Wah et al., 2011*)—served as source domain data, and Stanford Dogs (*Khosla et al., 2011*), MIT Indoor (*Quattoni & Torralba, 2009*), Caltech 256-30, Caltech 256-60 (*Griffin, Holub & Perona, 2007*), Aircraft (*Maji et al., 2013*), UCF-101 (*Bilen et al., 2016*), and Omniglot (*Lake, Salakhutdinov & Tenenbaum, 2015*) datasets were used as the target domain data, as shown in Table 1. We used ResNet50 pre-trained on ImageNet and CIFAR-100 datasets as the two source domain models for fine-tuning, respectively, whereby the classification accuracy of the model pre-trained on ImageNet was 75.15% (provided by the Torchvision library) and that of the model pre-trained on CIFAR-100 was 84.55%.

**Implementation details:** All experiments were conducted using the Pytorch framework on NVIDIA 3090 GPUs, utilizing SGD as the optimizer during training, with the weight decay and momentum set to 0.0005 and 0.9, respectively, and the initial learning rate and batch size set to 0.01 and 64, respectively. The number of fine-tuning epochs for evaluating each individual in our approach was set to 5. For the final fine-tuning, the number of epochs was set to 110, and learning rate decay was performed every 30 epochs. Each experiment was repeated five times to obtain the average classification accuracy.

## Parameter sensitivity analysis and multi-source domain model selection analysis

In AMCF, the selection of the source domains and the values of hyperparameters are most likely to affect the fine-tuning performance. Therefore, in this section, the analysis mainly focuses on these two factors. Firstly, regarding the selection of source domains, we conducted separate analyses and discussions on the performance of fine-tuning for target tasks using a single source domain (ImageNet), two source domains (ImageNet, CIFAR-100), and three source domains (ImageNet, CIFAR-100, CUB-200).

**Table 2 Comparative analysis of fine-tuning results of different selection of source domain models.** The best results are shown in bold.

| Source domains | MIT indoor | Stanford Dogs | Caltech256-30 | Caltech256-60 | Aircraft | UCF-101 | Omniglot |
|---|---|---|---|---|---|---|---|
| One source domain | 76.64% | 79.02% | 77.53% | 82.57% | 56.10% | 76.83% | 87.21% |
| Two source domains | 79.10% | **86.30%** | **82.91%** | 86.05% | **62.31%** | **80.54%** | **87.76%** |
| Three source domains | **79.25%** | 85.23% | 82.50% | **86.38%** | 61.14% | 79.96% | 86.87% |

Table 2 shows the classification accuracy of the proposed AMCF on two source domain models (ImageNet, CIFAR-100) and three source domain models (ImageNet, CIFAR-100, CUB-200), along with the classification accuracy achieved through full fine-tuning using only one source domain model (ImageNet). It can be seen from Table 2 that the classification accuracy obtained by AMCF is significantly higher than that obtained by a single source domain, whether it is two source domains or three source domains. In addition, the fine-tuning results obtained from using three source domains are worse than those obtained from using two source domains on five out of seven target tasks. This is because the source domain dataset CUB-200 is very dissimilar to these five target datasets, causing the source domain knowledge or features to be unsuitable for the target domain, thereby negatively impacting the model performance. However, in some target tasks, such as MIT Indoor and Caltech256-30, the results using three source domains are even better than using two source domains. Although the performance of two source domains and three source domains is similar, considering the computational complexity, using two source domains can achieve better results than three source domains, and the computational complexity is also lower. Therefore, in the subsequent experiments of this article, AMCF will use ImageNet and CIFAR-100 as the source domains.

The AMCF contains multiple hyperparameters such as the number of the population iterations $G$, population size $n$, learning factors $c_1$ and $c_2$, the number $maxEpoch$ of epochs for evaluating each individual, and $\beta$ in the loss function; where the values of $G$, $n$, $c_1$, and $c_2$ are referenced from previous studies (*Nagae et al., 2022*; *Eberhart & Shi, 2000*), and the number maxEpoch of epochs and the hyperparameter $\beta$ for evaluating each individual are new parameters introduced for this specific problem. In order to verify the impact of different hyperparameters on AMCF performance, we will compare and analyze the different values of $maxEpoch$ and $\beta$ in Table 3.

In this experiment, $\beta$ is set to 0, 0.1, 0.2, and 0.5, and $maxEpoch$ is set to 1, 3, 5, and 10, respectively. The value of $\beta$ determines the contribution ratio of the difference loss to the total loss, and the value of $maxEpoch$ represents the number of training epochs for evaluating each individual. Table 3 shows the performance of AMCF under different $\beta$ and $maxEpoch$ values. It can be seen from Table 3 that when $\beta = 0.2$, the best fine-tuning results can be obtained, and $maxEpoch$ is greater than or equal to 5, the performance of AMCF is better. In the following experiments, AMCF will use the parameter settings $\beta = 0.2$ and $maxEpoch = 5$.

**Table 3 Comparative analysis of fine-tuning results under different values of hyperparameters $\beta$ and *maxEpoch*.** The best results are shown in bold.

| $\beta$ | 0 | 0.1 | 0.2 | 0.5 |
|---|---|---|---|---|
| MIT indoors | 76.26% | 78.05% | **79.10%** | 78.50% |
| Stanford dogs | 84.97% | 85.37% | **86.30%** | 85.64% |
| Caltech256-30 | 78.82% | 81.58% | **82.91%** | 82.59% |
| Caltech256-60 | 83.45% | 85.52% | **86.05%** | 84.03% |
| *maxEpoch* | 1 | 3 | 5 | 10 |
| MIT indoors | 75.74% | 74.32% | **79.10%** | 78.95% |
| Stanford dogs | 83.90% | 85.68% | **86.30%** | 85.64% |
| Caltech256-30 | 79.53% | 82.01% | **82.91%** | 82.24% |
| Caltech256-60 | 80.76% | 85.41% | 86.05% | **86.23%** |

## Effectiveness analysis of the adaptive multi-source domain layer selection strategy

In this section, we analyze the fine-tuning performance of different fine-tuning layer selection strategies to verify the effectiveness of the proposed adaptive multi-source domain layer selection strategy (AMLS).

In this experiment, we fine-tuned the model using random selection, fine-tuning 20 layers, full fine-tuning, and AMLS. All four strategies use the MC-loss function. In AMLS, the population size $n$ was set to 10, the maximum number of population iterations $G$ was set to 5, and the number *maxEpoch* of epochs for each individual in assessing fitness was set to 5. In the process of individual speed updating, the learning factors $c_1$ and $c_2$ were set to 1.5, and the inertia weights, $\alpha$, were used in a linearly decreasing manner while remaining in the $\alpha \in [0.7, 1.4]$ range. In the experiments with the random selection strategy, the number of layers fine-tuned on all source domain models is a random number generated between 1 and 50. For example, (10, 30) means that the final 10 layers of the first source domain model are fine-tuned, while the final 30 layers of the second source domain model are fine-tuned. Fine-tuning 20 layers means fine-tuning the final 20 layers of all source domain models. Full fine-tuning refers to fine-tuning all layers. Based on the layer fine-tuning schemes obtained from the above strategies, we conducted 110 fine-tuning epochs on the source domain models and evaluated its classification accuracy on the seven image classification datasets.

Table 4 shows the classification accuracy of the four selection strategies when applied to the test sets derived from all datasets. It is evident that the AMLS achieved superior fine-tuning results on all datasets. Moreover, the layer fine-tuning result yielded by the random selection strategy is 6% lower than the AMLS on average, highlighting the importance of accurately selecting the fine-tuning layer. Although making an inappropriate selection can greatly reduce the model's performance during the fine-tuning process, the AMLS can accurately identify appropriate layers for fine-tuning in multiple source domain models, effectively enhancing the model's performance on the target task. Figure 4 shows the number of fine-tuned layers obtained by the AMLS on two source domain models. Source domain model CIFAR-100 was trained on the CIFAR-100 dataset, while source

**Table 4  Classification accuracy achieved using random selection, fine-tune 20 layers, full fine-tune, and the proposed AMLS.** The best results are shown in bold.

| Selection strategy | MIT indoor | Stanford Dogs | Caltech256-30 | Caltech256-60 | Aircraft | UCF-101 | Omniglot |
|---|---|---|---|---|---|---|---|
| Random selection | 77.91% | 81.32% | 77.59% | 79.02% | 49.50% | 72.43% | 85.15% |
| Fine-tune 20 layers | 77.98% | 81.68% | 78.98% | 82.89% | 61.77% | 80.07% | 85.22% |
| Full fine-tune | 78.35% | 81.34% | 78.84% | 82.75% | 60.21% | 79.96% | 86.09% |
| **AMLS** | **79.10%** | **86.30%** | **82.91%** | **86.05%** | **62.31%** | **80.54%** | **87.76%** |

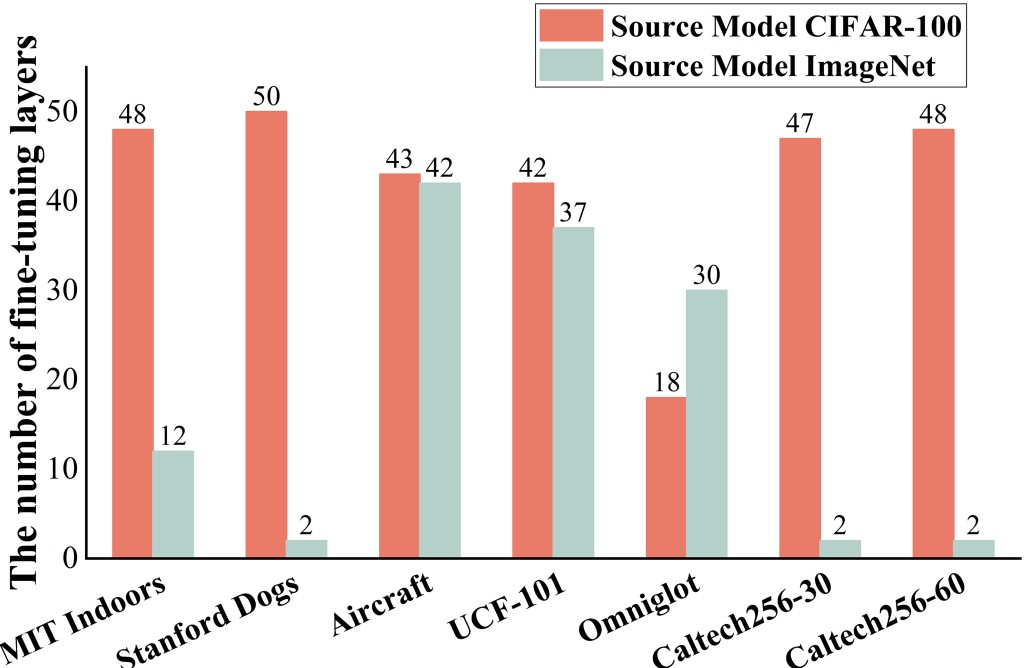

**Figure 4  The number of fine-tuning layers obtained *via* each source domain model using the AMLS.**

domain model ImageNet was trained on the ImageNet dataset. As shown in Fig. 4, we observed that in most datasets, the number of layers fine-tuned from the source domain model ImageNet is fewer than the model CIFAR-100. This suggests that the variance in data distribution between ImageNet and most target tasks is smaller compared to CIFAR-100. However, on Omniglot, the model fine-tuned from ImageNet has adjusted more layers than CIFAR-100, indicating a relatively larger difference in data distribution between ImageNet and Omniglot. In addition, the optimal number of fine-tuning layers varies significantly for different target tasks within the same source domain. This indicates that the optimal number of fine-tuning layers varies depending on the target task.

In summary, the experimental results shown in Table 4 and Fig. 4 indicate that the selection of fine-tuning layers in the source domain models plays a crucial role in extracting key features for the target task. The AMLS can accurately identify the fine-tuning layers suitable for the target task, thereby enabling the model to extract more crucial feature information.

**Table 5  Fine-tuning results of different loss functions on multiple source domain models.** The best results are shown in bold.

| Loss function | MIT indoor | Stanford dogs | Caltech256-30 | Caltech256-60 | Aircraft | UCF-101 | Omniglot |
|---|---|---|---|---|---|---|---|
| (CE-loss, Model CIFAR-100) | 70.74% | 75.86% | 67.91% | 77.16% | 51.45% | 76.25% | 86.59% |
| (CE-loss, Model ImageNet) | 76.19% | 84.79% | 82.59% | 84.65% | 54.39% | 76.25% | 87.30% |
| (CE-loss, Two Models) | 78.95% | 85.31% | 81.03% | 84.86% | 57.03% | 78.13% | 87.53% |
| **(MC-loss, Two Models)** | **79.10%** | **86.30%** | **82.91%** | **86.05%** | **62.31%** | **80.54%** | **87.76%** |

## Effectiveness analysis of the multi-source collaborative loss function

In this section, we report on the comparative experiments that were conducted to verify the fine-tuning effectiveness of the proposed MC-loss function when applied to the target task. For this purpose, a comparison with the cross-entropy loss function was conducted on both single-source domain and multi-source domain models using four sets of experiments. In all experiments, the source domain models adopted the same fine-tuning and freezing scheme (shown in Fig. 4), while the hyperparameter value $\beta$ was set to 0.2, and each set of experiments was trained for 110 epochs. The classification accuracy on the target task was recorded for each experiment, the details of which are outlined below.

(CE-loss, Model CIFAR-100): Fine-tuning the source domain model CIFAR-100 on all target datasets using the cross-entropy loss function.

(CE-loss, Model ImageNet): Fine-tuning the source domain model ImageNet on all target datasets using the cross-entropy loss function.

(CE-loss, Two models): The proposed AMCF applied for fine-tuning source domain model CIFAR-100 and source domain model ImageNet using the cross-entropy loss function.

(MC-loss, Two models): The proposed AMCF is applied, which involves fine-tuning source domain model CIFAR-100 and source domain model ImageNet using the multi-source domain collaborative loss function.

Table 5 shows the classification accuracy achieved by the aforementioned four fine-tuning combinations on seven datasets, confirming that the AMCF is superior to the single-source domain fine-tuning method. These results suggest that a single source domain model can only extract limited key features for the target task, whereas collaborative fine-tuning with multiple source domains can extract more key features, leading to a significant improvement in the model performance on the target task.

The experimental results show that the MC-loss function enhances the accuracy of extracting key features in the target task by adapting each source domain model to the target task while minimizing the discrepancies between the output features of different source domain models. Ultimately, this significantly enhances the fine-tuning performance of the model on the target task.

## Comparison of the adaptive multi-source collaborative fine-tuning framework with the popular single-source domain fine-tuning methods

In this section, we compare and analyze the AMCF with several popular fine-tuning methods, including Train-From-Scratch, Standard Fine-Tuning (Standard-FT), L2-SP,

**Table 6 Comparative analysis of Top-1 classification accuracy between adaptive multi-source domain collaborative fine-tuning framework and single-source domain fine-tuning methods.** The best results are shown in bold.

| Method | MIT indoor | Stanford dogs | Caltech256-30 | Caltech256-60 | Aircraft | UCF-101 | Omniglot |
|---|---|---|---|---|---|---|---|
| Train-From-Scratch | 40.82% | 42.45% | 25.41% | 47.55% | 12.12% | 43.61% | 84.82% |
| Standard-FT (ImageNet) | 76.64% | 79.02% | 77.53% | 82.57% | 56.10% | 76.83% | 87.21% |
| Standard-FT (CIFAR-100) | 72.38% | 75.85% | 68.53% | 77.20% | 55.59% | 76.53% | 87.32% |
| L2-SP | 76.41% | 79.69% | 79.33% | 82.89% | 56.52% | 74.33% | 86.92% |
| Child-Tuning | 77.83% | 81.13% | 80.19% | 83.63% | 55.92% | 77.40% | 87.32% |
| AdaFilter | 77.53% | 82.44% | 80.62% | 84.31% | 55.41% | 76.99% | 87.46% |
| ALS | 76.64% | 83.34% | 80.93% | 84.21% | 56.04% | 75.78% | 87.09% |
| Auto-RGN | 77.46% | 83.21% | 80.44% | 82.98% | 54.18% | 76.20% | 87.43% |
| OFT | 77.68% | 83.44% | 79.49% | 83.82% | 55.89% | 74.21% | 84.02% |
| **Ours** | **79.10%** | **86.30%** | **82.91%** | **86.05%** | **62.31%** | **80.54%** | **87.76%** |

Child-Tuning, AdaFilter, ALS, Auto-RGN, and OFT. Standard-FT (ImageNet) and Standard-FT (CIFAR-100) represent standard fine-tuning using the source domain models ImageNet and CIFAR-100, respectively. The effectiveness of AMCF is validated by applying all alternatives to seven public datasets. All methods were iterated 110 epochs and each method was applied five times to obtain the average classification accuracy and eliminate the influence of randomness. The experimental results are presented in Table 6 and Fig. 5, whereby those related to the L2-SP and AdaFilter methods were sourced from published works (*Xuhong, Grandvalet & Davoine, 2018*; *Guo et al., 2020*).

As can be seen from the Table 6, the proposed AMCF achieved the highest accuracy on all datasets. The Train-From-Scratch method achieved the lowest accuracy as it trained the target task using randomly initialized weight parameters, without employing any fine-tuning techniques. Standard Fine-Tuning, L2-SP, Child-Tuning, AdaFilter, ALS, Auto-RGN, and OFT methods are single-source domain fine-tuning techniques and utilize the source domain model for fine-tuning. Accordingly, they greatly enhance performance compared to the Train-From-Scratch approach. However, the performance of single-source domain fine-tuning methods is limited when there is a significant difference in data distribution between the source and the target domain, as the key features extracted from a single source domain model on the target task are limited. In contrast, the AMCF combines rich features extracted from multiple source domain models, making the model's predictions on the target task more accurate and effectively improving the model performance. For example, in Table 6, the performance of the source domain model CIFAR-100 after standard fine-tuning on six target tasks (excluding Omniglot) is lower than the performance obtained by the source domain model ImageNet. This indicates that the data distribution difference between the source domain CIFAR-100 and the six target domains is larger when compared to that of the source domain ImageNet, while the data distribution difference of the source domain ImageNet is relatively smaller. This conclusion is consistent with the findings regarding the number of fine-tuning layers in Fig. 4. When using AMCF to combine two source domain models, the accuracy obtained on all target tasks is significantly better than the results of fine-tuning a single source domain. This also indicates that, even in

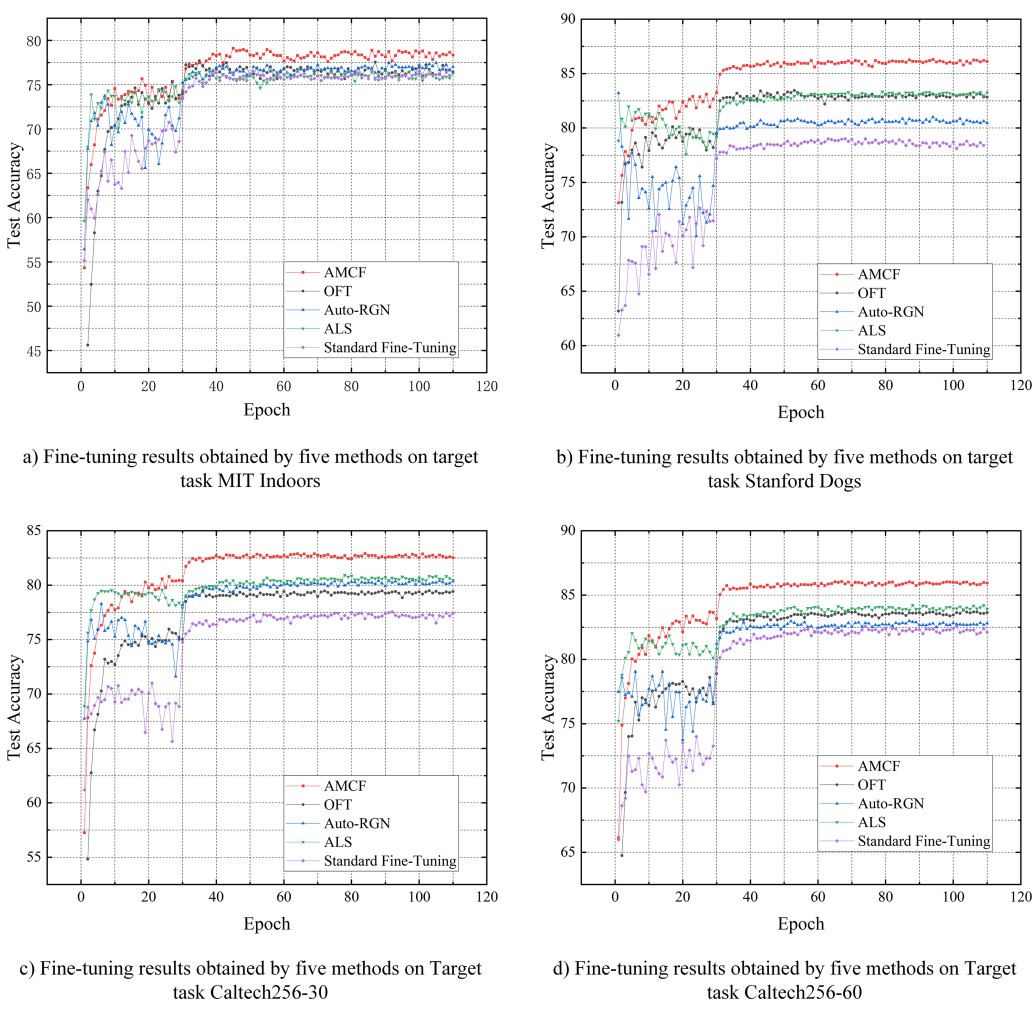

a) Fine-tuning results obtained by five methods on target task MIT Indoors

b) Fine-tuning results obtained by five methods on target task Stanford Dogs

c) Fine-tuning results obtained by five methods on Target task Caltech256-30

d) Fine-tuning results obtained by five methods on Target task Caltech256-60

**Figure 5** **Test accuracy curve of the proposed method, OFT, Auto-RGN, ALS, and Standard Fine-Tuning baseline when applied to MIT Indoors, Stanford Dogs, Caltech256-30, and Caltech256-60 datasets.**

instances when there is a significant distribution difference between a single source domain and a target domain, AMCF can still enhance the model's feature extraction capability. However, even though the AMCF can enhance the model's feature extraction ability on the target task when there is a significant difference in data distribution between the source and target domain, the performance improvement obtained through fine-tuning is still limited when the data distribution differences between all source domain and target domain are too large or completely unrelated. For example, the target task Omniglot achieved only about a 2% accuracy improvement compared to Training-From-Scratch when using standard fine-tuning on the source domain models ImageNet and CIFAR-100. This indicates that the data distribution of Omniglot is very different from that of both source domains. And thus, AMCF only achieved an improvement of around 0.3% compared to other single-source domain fine-tuning methods. Figure 5 shows the test accuracy curves of our method, OFT,

Auto-RGN, ALS, and Standard Fine-Tuning (source ImageNet) baseline when applied to the Stanford Dogs, MIT Indoors, Caltech256-30, and Caltech256-60 datasets. The red curve represents the fine-tuning results of the proposed AMCF. It is evident that the proposed method significantly improves fine-tuning performance compared to other fine-tuning methods on all datasets. For example, after 5 epochs, our method achieves classification accuracies of 79.80%, 71.11%, 75.91%, and 80.03%, while the standard fine-tuning baseline method only achieves 67.73%, 62.61%, 69.66%, and 71.28%. Therefore, our method can match or even surpass the accuracy level of the standard fine-tuning baseline approach in fewer epochs, significantly reducing training time for new tasks, making it more suited for practical applications.

These experimental results demonstrate that the proposed AMCF effectively extracts a greater number of key features in the target task by fine-tuning multiple source domain models. Compared to current single-source domain fine-tuning methods, it greatly enhances the fine-tuning performance, while achieving competitive results within fewer training epochs, as shown in Fig. 5.

### Experimental findings

Based on the experimental results, the following conclusions can be reached: (1) multi-source collaborative domain fine-tuning is more beneficial for learning effective feature information in the target task, leading to a significant improvement in model performance; (2) the selection of fine-tuning layers in the source domain model is crucial for achieving optimal fine-tuning performance, and this selection varies depending on the target task. The proposed AMLS can accurately identify the appropriate layers for the target task.

While this method improves the model performance on the target task compared to single-source domain fine-tuning strategy, training multiple source domain models can lead to slower inference speed. When there are $N$ source domain models, the number of parameters required for training will be approximately $N$ times greater than that in single-source domain fine-tuning approach. In the future, this issue will be addressed by reducing the number of fine-tuning parameters or by incorporating adapter fine-tuning. Additionally, it is important to note that the AMCF method only modifies the output layer and loss function components of multi-source domain models. Fine-tuning is performed on a per-layer basis, making it applicable to various deep neural network model architectures.

## CONCLUSION

In order to mitigate the difficulty in extracting effective features through single-source domain fine-tuning when there is a significant difference in data distribution between the source domain and the target domain, we proposed a transfer learning framework called adaptive multi-source domain collaborative fine-tuning (AMCF). The framework combines multiple source domain models for collaborative fine-tuning, extracting more effective features for the target task compared to single-source domain fine-tuning methods, thereby significantly improving the fine-tuning performance. In the proposed method, an adaptive multi-source domain layer selection strategy is designed to dynamically provide appropriate layer fine-tuning schemes for the target task on multiple source domain

models, enabling the model to extract more crucial feature information. Based on this strategy, a multi-source domain collaborative loss function is designed to adapt each source domain model to the target task. The strategy aims to minimize the differences between the output features of different source domain models, thereby improving the accuracy of the key features extracted by the model in the target task. Experimental results show that compared with the most widely used single source domain fine-tuning methods, the proposed AMCF extracts more effective key feature information of the target task, thereby significantly improving the performance of model fine-tuning.

However, since the AMCF combines multiple source domain models for collaborative fine-tuning, its parameter scale is extremely large, making it difficult to execute this approach on devices with limited computational resources. Therefore, our future work will primarily focus on exploring multi-source domain collaborative fine-tuning methods that can be adopted in such contexts.

## ACKNOWLEDGEMENTS

We truly thank the reviewers for the pertinent comments.

### Funding

This work is supported by the National Natural Science Foundation of China (62162012), the Guizhou Provincial Science and Technology Projects (QKHJCZK2022YB195, QKHJCZK2023YB143, QKHPTRCZCKJ2021007), the Natural Science Research Project of Education Department of Guizhou Province (QJJ2023061, QJJ2023012, QJJ2022015), the Guizhou province pattern recognition and intelligent system key laboratory open subject (GZMUKL2022KF01, GZMUKL2022KF05), the High-Level Innovative Talent Project of Guizhou Province (QKHPTRC-GCC2023027), and the Youth Science and Technology Talent Growth Project of Guizhou Province (QJHKY2022319). The funders had no role in study design, data collection and analysis, decision to publish, or preparation of the manuscript.

### Grant Disclosures

The following grant information was disclosed by the authors:
The National Natural Science Foundation of China: 62162012.
The Guizhou Provincial Science and Technology Projects: QKHJCZK2022YB195, QKHJCZK2023YB143, QKHPTRCZCKJ2021007.
The Natural Science Research Project of Education Department of Guizhou Province: QJJ2023061, QJJ2023012, QJJ2022015.
Guizhou province pattern recognition and intelligent system key laboratory open subject: GZMUKL2022KF01, GZMUKL2022KF05.
The High-Level Innovative Talent Project of Guizhou Province: QKHPTRC-GCC2023027.
The Youth Science and Technology Talent Growth Project of Guizhou Province: QJHKY2022319.

## Competing Interests

The authors declare there are no competing interests.

## Author Contributions

- Le Feng conceived and designed the experiments, performed the experiments, analyzed the data, performed the computation work, prepared figures and/or tables, authored or reviewed drafts of the article, and approved the final draft.
- Yuan Yang conceived and designed the experiments, performed the experiments, analyzed the data, performed the computation work, prepared figures and/or tables, authored or reviewed drafts of the article, and approved the final draft.
- Mian Tan conceived and designed the experiments, analyzed the data, authored or reviewed drafts of the article, and approved the final draft.
- Taotao Zeng performed the experiments, analyzed the data, performed the computation work, authored or reviewed drafts of the article, and approved the final draft.
- Huachun Tang performed the experiments, analyzed the data, prepared figures and/or tables, authored or reviewed drafts of the article, and approved the final draft.
- Zhiling Li performed the experiments, prepared figures and/or tables, authored or reviewed drafts of the article, and approved the final draft.
- Zhizhong Niu analyzed the data, prepared figures and/or tables, authored or reviewed drafts of the article, and approved the final draft.
- Fujian Feng conceived and designed the experiments, performed the experiments, analyzed the data, prepared figures and/or tables, and approved the final draft.

## Data Availability

The CIFAR100 dataset is available at https://www.cs.toronto.edu/~kriz/cifar.html and at Kaggle: https://www.kaggle.com/datasets/aymenboulila2/cifar100.

The CUB-200 dataset is available at https://www.vision.caltech.edu/datasets/cub_200_2011 and at Kaggle: https://www.kaggle.com/datasets/veeralakrishna/200-bird-species-with-11788-images.

The MIT Indoor dataset is available at https://web.mit.edu/torralba/www/indoor.html and at Kaggle: https://www.kaggle.com/datasets/itsahmad/indoor-scenes-cvpr-2019.

The Stanford Dogs dataset is available at http://vision.stanford.edu/aditya86/ImageNetDogs and at Kaggle: https://www.kaggle.com/datasets/jessicali9530/stanford-dogs-dataset.

The Caltech 256-30 and Caltech 256-60 is available at https://data.caltech.edu/records/nyy15-4j048 and at Kaggle: https://www.kaggle.com/datasets/mmoreaux/caltech256.

The FGVC-Aircraft Benchmark is available at https://www.robots.ox.ac.uk/~vgg/data/fgvc-aircraft and at Kaggle: https://www.kaggle.com/datasets/seryouxblaster764/fgvc-aircraft.

The UCF101 Dynamic Images dataset is available at https://www.robots.ox.ac.uk/~vgg/decathlon/#download.

The Omniglot dataset is available at https://www.robots.ox.ac.uk/~vgg/decathlon.

The code is available at GitHub and Zenodo:
- https://github.com/fengledl/AMCF
- fengledl. (2024). fengledl/AMCF: v1.0.0 (v1.0.0). Zenodo. https://doi.org/10.5281/zenodo.10827073.

### Supplemental Information

Supplemental information for this article can be found online at http://dx.doi.org/10.7717/peerj-cs.2107#supplemental-information.

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
