# Peer review of "Adaptive multi-source domain collaborative fine-tuning for transfer learning"

_PeerJ Computer Science, doi:10.7717/peerj-cs.2107_

## Round 0.1 · original submission · Major Revisions

Dear authors,

You are advised to critically respond to all comments point by point when preparing a new version of the manuscript and while preparing for the rebuttal letter. Please address all the comments/suggestions provided by the reviewers.

Kind regards,
PCoelho

**Language Note:** The review process has identified that the English language must be improved. PeerJ can provide language editing services - please contact us at copyediting@peerj.com for pricing (be sure to provide your manuscript number and title). Alternatively, you should make your own arrangements to improve the language quality and provide details in your response letter. – PeerJ Staff

Reviewer 1 ·

Basic reporting

no comment

Experimental design

no comment

Validity of the findings

no comment

Additional comments

In this manuscript, the authors propose an adaptive multi-source domain collaborative fine-tuning
framework which unifies multiple source-domain models into a single framework and fine-tunes them on the target task, thereby improving the model’s ability to extract the key features of the target task. This is a very interesting work. However, the authors should consider the following issues before resubmitting:
(i) The authors should show the complexity of the Algorithm 1.
(ii) If possible, then consider more current works in case of comparison.
(iii) Please show the impact by analyzing the results of more (not only 2) source domain models in the system.

Cite this review as

·

Basic reporting

The article generally demonstrates clear and professional English, adhering to standards. Literature references and field background are sufficiently provided, enhancing context. The article maintains a professional structure with well-organized figures and tables. However, there is no explicit mention of raw data sharing. The document is self-contained, presenting relevant results aligned with hypotheses. While clear definitions of terms and theorems are provided, detailed proofs are not explicitly mentioned, suggesting a need for clarification in this aspect. Overall, the article adheres well to basic reporting standards, with minor improvements recommended.

Experimental design

The article conforms to experimental design standards. The research is original and aligns with the journal's scope. The research question is well-defined and relevant, and it addresses a knowledge gap. The investigation is rigorous, meeting high technical and ethical standards.

Validity of the findings

The article requires improvement in evaluating impact and novelty. Encourage meaningful replication with a clear rationale and benefits to the literature. Ensure robust, statistically sound, and controlled underlying data. Conclusions should be well-stated, linked to the original research question, and limited to supporting results.

Additional comments

The manuscript demonstrates a commendable effort in proposing an adaptive multi-source domain collaborative fine-tuning method. However, attention is needed to providing clearer explanations for certain technical aspects. Additionally, refining the language for better readability is advised. The experimental design and results are robust, but the impact and novelty of the findings could be emphasized further. Overall, with minor revisions to enhance clarity and emphasize novelty, the paper has the potential for acceptance.

Reviewer 3 ·

Basic reporting

The document exhibits a clear and unambiguous expression, with technically accurate text employed.
Sufficient background is provided with relevant literature references. The contribution of proposed method is clearly stated in the paper.

Experimental design

The experimental design is well presented to fulfill the research gap and solve the problem stated. Rigorous investigation is performed with sufficient quantitative experiments (multiple models and multiple datasets). For further improvement, the authors can consider the following comments:

1. Providing additional details on how the combination of cross-entropy loss and difference loss can be used to treat the models equally, which finally would enhance the classification accuracy.
2. It would be preferable to include a description of how the hyperparameter value is determined in the manuscript (epoch, in PSO such as no. of population iteration, learning factors etc, hyperparameters in loss function measurement)

Validity of the findings

This manuscript presents sufficient quantitative experiments (multiple models and multiple datasets) with clear result and discussion. The analysis of effectiveness for the adaptive multi-source domain layer selection strategy and multi-source collaborative loss function is thoroughly presented via promising experimental result and comparative analysis. The conclusion is also well stated, linked to the stated problem statement and the findings supports the contribution of the proposed work.

Additional comments

1. To enhance readability, it is suggested to position Figure 2 after its first reference in the text, namely after the first paragraph in the section titled "Overview of the Proposed Framework" or at the bottom of page 5/16.
2. Same comment with (1) for Table of Algorithm 1, Table 2 , Figure 3 and Figure 4. To enhance readability, it is suggested to position these figures/table after its first reference in the text.

Cite this review as

Reviewer 4 ·

Basic reporting

The article generally employs clear language, although there is room for slight improvement to enhance readability. Please find in comments

Experimental design

No comment

Validity of the findings

No comment

Additional comments

Summary of the paper

Authors proposed that fine-tuning collaboratively using multiple pre-trained models to a downstream task can help learn better features and consequently obtain better performance compared to using a single source pre-trained model (which could suffer when the data distribution between source and target is large). In that regard, authors have proposed, a) Algorithm to adaptively select layers that need to be fine-tuned or frozen, b) multi-collaborative loss function to reduce the discrepancy between the different fine-tuned models.

The number of benchmark datasets and method comparisons used to demonstrate the findings is commendable.

Major comments

1. It is stated in the paper that multi-collaborative fine-tuning can still perform well when there is large difference in data distributions between the source and target domain. However, I believe that this was not established in the findings.
a. Through using multiple pre-trained models, it looks like you the chances of pre-trained models to be more informative of the downstream target task is increased compared to when using just one pre-trained model. For example, source model A might not be informative (or data distribution difference is large), but source model B might be informative. So, to my understanding, the proposed method is aggregating this information to help learn better features and improve model performance. However, in a scenario where all pre-trained models are not very informative about the downstream task, I am expecting that the proposed method might also suffer from this large difference in distribution. For example, the performance improvement for “Omniglot” dataset was minimal in Table 4. Can the authors comment on this?
b. I think Table 4 can be used as an opportunity to relate the findings to the claim that is being made about the difference in the data distribution.

2. In Table 4, I think it would be valuable to give some insights to the readers on how the performance varies when all layers are fine-tuned (skipping the AMLS) but using multi-collaborative loss function. So that the value of AMLS can be understood more clearly.

3. For Figure 3, I think this would be an opportunity to discuss, why ImageNet had less layers to fine-tune compared to CIFAR100 for most of the datasets, whereas for just Omniglot, ImageNet # layers to fine-tune is higher. And probably, the difference in data distribution between a source dataset like ImageNet and target dataset like Omniglot can be discussed using the # of layers need to fine-tune. For example, does higher layers to fine-tune indicate large difference in distribution? And I think it would be useful if this discussion is connected to the findings in Table 4, on why there isn’t much performance improvement for Omniglot dataset.

4. Table 4 results clearly indicate that the proposed method outperforms other single source fine-tuning methods. However, it is not clear if these are fine-tuned from CIFAR100 or ImageNet. For example, standard fine-tuning, it would be good to see how fine-tuning from both pre-training datasets look.

5. For Table 2, more explanation for “Random selection” would be very helpful. Were they just randomly chosen, ist that all? Also, adding a comparison with fine-tuning all layers could be helpful here.

6. Some explanation on why 5 was chosen as Maxepoch, what difference would it make if 3 or 10 was chosen could be helpful.

Minor comments

1. Experimental findings section could be a place for discussion, a) elaborating on the limitations that were already mentioned like high training and inference cost and potential solutions, b) how the method is independent of the pre-trained model architectures, and different source pre-trained models could have different architectures?
2. Instead of top/ bottom – initial and final layers. Improving this explanation could help readability. On how, the algorithm identifies the X number of layers to fine-tune and the final X layers of the architecture are fine-tuned.
3. Please check ImageNet reference in introduction.
4. For Figure 3, it would be helpful if you can add which model is CIFAR100 and ImageNet in the legend. Same with Table 3.
5. Line 50 – I think you are referring to Figure 1 here which wasn’t clear, if you are suggesting to review the reference you provided. For clarity, I think it is better if you state as shown in Figure 1 (adopted from the citation).

Cite this review as

---

## Round 0.2 · Minor Revisions

Dear authors,

Thanks a lot for your efforts to improve the manuscript.

Nevertheless, some concerns are still remaining that need to be addressed.

Like before, you are advised to critically respond to the remaining comments point by point when preparing a new version of the manuscript and while preparing for the rebuttal letter.

Kind regards,
PCoelho

Reviewer 3 ·

Basic reporting

The English language in manuscript requires additional proofreading. Certain statements are lengthy and difficult to comprehend.For example, in page 2, second paragraph, most of the sentences are 3–4 lines long, causing difficulties in understanding the intended meaning.

The manuscript should include a literature review related to the fine-tuning of multi-source domains as well. These papers only reviewed methods suitable for fine-tuning a single source.

Experimental design

Please elaborate more on what it means by "significant difference in data distribution between the source and the target domain." It is unclear why this problem necessitates a multi-source domain with fine tuning.

In contrast, the proposed method is described with commendable detail, covering all necessary aspects required for understanding and replication.

Validity of the findings

The author should provide a rationale for the better performance of AMCF when using two source domains, as compared to using only one source domain or three source domains.

Please provide information on how the number of fine-tuning layers acquired from each source domain model using the AMLS affects the final outcome. Does a greater value indicate a better outcome? What are the findings derived from Figure 4?

Cite this review as

Reviewer 4 ·

Basic reporting

No comment

Experimental design

no comment

Validity of the findings

no comment

Additional comments

I would like to thank the authors for addressing my comments or concerns. I have no major comments. A few minor ones are listed below.

1. It is good to see that AMLS achieves better performance than full fine-tuning demonstrating the effectiveness of identifying the layers to fine-tune. I think it would be nice to make this clear in the table. For which of these experiments MC-loss was used.
2. Looks like a third pre-trained dataset was chosen CUB-200, and looks like outperforms single source model but is comparable to two source domains. I think this begs the question if a) there is a systematic way to select the source models, b) when does adding more and more source models really help. These are challenging ones but some discussion on this could be helpful I think. For example, can we say something like, adding the three source domain did not really deteriorate the performance compared to the two source, it increased the computation complexity of course. But, when tested on different target datasets, could be beneficial, while two source mode might perform worse.
3. Check line 223, I think it either has to be effective features for the target domain or from the source domain.

Cite this review as

---

## Round 0.3 · accepted · Accept

Dear authors, we are pleased to verify that you meet the reviewer's valuable feedback to improve your research.

Thank you for considering PeerJ Computer Science and submitting your work.

Reviewer 3 ·

Basic reporting

No Comment

Experimental design

No Comment

Validity of the findings

No Comment

Additional comments

All the comments have been addressed accordingly.

Cite this review as